# Sphingosine-1-phosphate activates LRRC8 volume-regulated anion channels through G$\beta\gamma$ signalling

Yulia Kostritskaia[1] , Sumaira Pervaiz[2] , Anna Klemmer[1] , Malte Klüssendorf[1] and Tobias Stauber[1,2]

[1] *Institute for Molecular Medicine, MSH Medical School Hamburg, Hamburg, Germany*
[2] *Institute of Chemistry and Biochemistry, Freie Universität Berlin, Berlin, Germany*

The peer review history is available in the Supporting Information section of this article (https://doi.org/10.1113/JP286665#support-information-section).

**Abstract figure legend** Sphingosine-1-phosphate (S1P) binds to Gi protein-coupled receptor S1PR1. Upon S1PR1 activation, G$\beta\gamma$ is released from the G$\alpha$i-G$\beta\gamma$ heterotrimer, allowing it to recruit phospholipase C$\beta$ (PLC$\beta$) to the plasma membrane. PLC$\beta$ cleaves phosphatidylinositol-4,5-bisphosphate (PIP$_2$) into inositol trisphosphate (IP$_3$) and diacylglycerol (DAG). DAG recruits and activates protein kinase D (PKD), which in turn may lead to leucin-rich repeat containing 8 (LRRC8)/volume-regulated anion channel (VRAC) activation. Created with BioRender.com.

 

**Yulia Kostritskaia** studied biophysics at the St Petersburg Polytechnic University and completed her Master's thesis in 2016 at the Institute of Cytology, where she investigated ion channel functions in neurodegenerative diseases. She conducted her PhD studies at the Forschungszentrum Jülich and received her doctorate from Heinrich Heine University Düsseldorf in 2020. Since 2021, she has been a research associate in Tobias Stauber's lab at the Medical School Hamburg. Her research focuses on the mechanism and regulation of ion transport processes. **Sumaira Pervaiz** completed her Master's degree in Biochemistry at the University of the Punjab, Pakistan. She then joined Tobias Stauber's lab at the Freie Universität Berlin, where she obtained her PhD. Her doctoral research focused on the subunit composition of the volume-regulated anion channel and its isosmotic activation, which she studied using biochemical and optical methods.

**Abstract** Volume-regulated anion channels (VRACs) formed by leucin-rich repeat containing 8 (LRRC8) proteins play a pivotal role in regulatory volume decrease by mediating the release of chloride and organic osmolytes. Apart from the regulation of cell volume, LRRC8/VRAC function underlies numerous physiological processes in vertebrate cells including membrane potential regulation, glutamate release and apoptosis. LRRC8/VRACs are also permeable to antibiotics and anti-cancer drugs, representing therefore important therapeutic targets. The activation mechanisms for LRRC8/VRACs are still unclear. Besides through osmotic cell swelling, LRRC8/VRACs can be activated by various stimuli under isovolumetric conditions. Sphingosine-1-phosphate (S1P), an important signalling lipid, which signals through a family of G protein-coupled receptors (GPCRs), has been reported to activate LRRC8/VRACs in several cell lines. Here, we measured inter-subunit Förster resonance energy transfer (FRET) and used whole-cell patch clamp electrophysiology to investigate S1P-induced LRRC8/VRAC activation. We systematically assessed the involvement of GPCRs and G protein-mediated signal transduction in channel activation. We found that S1P-induced channel activation is mediated by S1PR1 in HeLa cells. Following the downstream signalling pathway of S1PR1 and using toxin-mediated inhibition of the associated G proteins, we showed that $G\beta\gamma$ dimers rather than $G\alpha i$ or $G\alpha q$ play a critical role in S1P-induced VRAC activation. We could also show that S1P causes protein kinase D (PKD) phosphorylation, suggesting that $G\beta\gamma$ recruits phospholipase $C\beta$ ($PLC\beta$) with the consequent PKD activation by diacylglycerol. Notably, S1P did not activate LRRC8/VRAC in HEK293 cells, but overexpression of $G\beta\gamma$-responsive $PLC\beta$ isoform could facilitate S1P-induced LRRC8/VRAC currents. We thus identified S1PR1-mediated $G\beta\gamma$-$PLC\beta$ signalling as a key mechanism underlying isosmotic LRRC8/VRAC activation.

(Received 2 April 2024; accepted after revision 15 October 2024; first published online 4 November 2024)
**Corresponding author** T. Stauber: Institute for Molecular Medicine, MSH Medical School Hamburg, Hamburg, Germany. Email: tobias.stauber@medicalschool-hamburg.de

**Key points**

- Leucin-rich repeat containing 8 (LRRC8) anion/osmolyte channels are involved in multiple physiological processes where they can be activated as volume-regulated anion channels (VRACs) by osmotic cell swelling or isovolumetric stimuli such as sphingosine-1-phosphate (S1P).
- In the present study, using pharmacological modulation and gene-depleted cells in patch clamp recording and optical monitoring of LRRC8 activity, we find that LRRC8/VRAC activation by S1P is mediated by the G protein-coupled receptor S1PR1 coupled to G proteins of the Gi family.
- The signal transduction to LRRC8/VRAC activation specifically involves phospholipase $C\beta$ activation by $\beta\gamma$ subunits of pertussis toxin-insensitive heteromeric Gi proteins.
- S1P-mediated and hypotonicity-induced LRRC8/VRAC activation pathways converge in protein kinase D activation.

# Introduction

Ion channels formed by heteromers of leucin-rich repeat containing 8 (LRRC8) family proteins are expressed ubiquitously in vertebrate cells where they are involved in numerous physiological processes. Their function as volume-regulated anion channels (VRACs) or volume-sensitive outwardly rectifying channels (VSORs) is central to the osmotic regulation of cell volume (Qiu et al., 2014; Voss et al., 2014). VRACs mediate the swelling-induced extrusion of chloride and organic osmolytes that subsequently drives water efflux allowing for regulatory volume decrease. Apart from volume regulation during osmotic stress, this function of LRRC8/VRACs has been implicated in isosmotic processes such as cell proliferation, migration and apoptosis (Chen, König, et al., 2019; Jentsch, 2016). LRRC8/VRAC anion currents affect the membrane potential, which was shown to play an important role in epithelial transport (Hoffmann et al., 2007),

myogenic differentiation (Chen, Becker, et al., 2019) and insulin secretion in pancreatic $\beta$-cells (Kang et al., 2018; Stuhlmann et al., 2018). LRRC8/VRACs mediate the transport of signalling molecules such as ATP, cGAMP, glutamate and aspartate, thereby facilitating communication between cells (Burow et al., 2015; Gaitán-Peñas et al., 2016; Lahey et al., 2020; Lutter et al., 2017; Mongin & Kimelberg, 2005; Schober et al., 2017; Yang et al., 2019; Zhou et al., 2020). Moreover, LRRC8/VRACs facilitate cellular uptake of certain xenobiotics such as cell dyes, antibiotics and anti-cancer drugs (Lee et al., 2014; Model et al., 2022; Planells-Cases et al., 2015; Strange et al., 2019).

Despite extensive research on the activation of LRRC8 channels and the implication of a variety of mechanisms, it remains unclear how they are activated under physiological conditions and whether their activation in hypotonic and isotonic conditions share the same mechanism (Bertelli et al., 2021; König & Stauber, 2019; Strange et al., 2019). Because major changes in extracellular osmolarity are rare under normal physiological conditions in vertebrates (Pedersen et al., 2015), isovolumetric stimuli may be more physiologically relevant for LRRC8/VRAC activation. Isosmotic LRRC8/VRAC activation by sphingosine-1-phosphate (S1P) was shown in the macrophage cell line RAW 264.7, where, together with concomitant ATP release, it establishes a functional link between sphingolipid and purinergic signalling (Burow et al., 2015). Subsequently, S1P was shown to activate VRAC in epithelial breast cancer and non-carcinogenic breast cell lines as well as in microglia (Chu et al., 2023; Furuya et al., 2021; Zahiri et al., 2021). S1P, which is produced by sphingosine kinase SphK1 or SphK2, is secreted by many cells and acts in an autocrine or paracrine manner (Maceyka et al., 2012). Extracellular S1P signals through a family of G protein-coupled receptors (GPCRs), S1PR1–S1PR5, which activate distinct heterotrimeric G proteins, Gi, Gq or G$_{12/13}$ (Hisano et al., 2012).

In the present study, we use Förster resonance energy transfer (FRET) and whole-cell patch clamp to monitor LRRC8/VRAC activity in response to S1P. We systematically investigated the S1P signalling pathway in HeLa cells by sequentially blocking individual steps. We found that S1P-induced isosmotic LRRC8/VRAC activation is mediated by G$\beta\gamma$ from G$\alpha$i-G$\beta\gamma$ heterotrimers, whereas hypotonicity-induced VRAC activation is not. However, we found that both S1P and hypotonic stimuli lead to the activation of protein kinase D (PKD) as shown by its phosphorylation. PKD activation represents therefore a meeting point for different signalling pathways leading to LRRC8/VRAC activation.

## Methods

### Cell culture and transfection

HeLa (RRID: CVCL_0030) and HEK293 (RRID: CVCL_0045) cells were obtained from the Leibniz Forschungsinstitut DSMZ (Braunschweig, Germany). The S1PR1 (S1P1/EDG1) knockout (KO) HeLa cell line was obtained from Abcam (Cambridge, UK) (catalog. no. ab265936; RRID: CVCL_B2EK). The LRRC8A KO HeLa cells (Yang et al., 2019) and the LRRC8 KO HEK293 cells depleted of all LRRC8 paralogues (Lutter et al., 2017) were kindly provided by Zhaozhu Qiu (Johns Hopkins University School of Medicine, Baltimore, MD, USA) and Thomas J. Jentsch (Leibniz-Institut für Molekulare Pharmakologie, Berlin, Germany), respectively. Cells were grown in Dulbecco's modified Eagle's medium supplemented with 10% fetal bovine serum, 1% penicillin–streptomycin and 1% glutamine (all from Pan-Biotech, Aidenbach, Germany) in plastic tissue culture flasks at 37°C with 5% $CO_2$ and 100% humidity atmosphere.

For the patch clamp experiments, HeLa cells were plated in 35 mm plastic dishes (Greiner GmbH, Pleidelsheim, Germany) and HEK293 cells were plated in 35 mm no 1.5 polymer coverslip bottom dishes (Ibidi, Gräfelfing, Germany) 24–36 h prior to current recording. Cells were transfected with cDNA encoding for phospholipase (PLC)$\beta$3 (Sino Biological, Beijing, China; catalog. no. HG18971-UT), S1PR1 or S1PR2. For the generation of S1PR1 and S1PR2 expression constructs, the cDNA was amplified by PCR on templates S1PR1- and S1PR2-Tango (Kroeze et al., 2015), a gift from Bryan Roth (Addgene, Watertown, MA, USA; plasmids #66496 and #66497), introducing an *EcoR*I cleavage site upstream of the start codon and a stop codon followed by an *Xho*I site at the 3′ end. After restriction digestion, the cDNA was inserted into pcDNA3.1(+) (Thermo Fisher Scientific, Waltham, MA, USA). Then, 0.25 µg of PLC$\beta$3, S1PR1 or S1PR2 cDNA was co-transfected with 1.25 µg of pEGFP-C1 (Clontech, Mountain View, CA, USA) using the Ca$_3$(PO$_4$)$_2$ technique and GFP-positive cells were selected for the recordings.

For FRET experiments, HeLa cells were plated in 35 mm glass bottom dishes. Cells were co-transfected with LRRC8A-Cerulean and LRRC8E-Venus (König et al., 2019) using FuGENE 6 (Promega, Madison, WI, USA) in accordance with the manufacturer's instructions. Next, 500 ng of each plasmid DNA was used and cells were imaged 24 h after transfection. For the experiments with blockers, all blockers were diluted in an isotonic solution containing (in mM): 150 NaCl, 6 KCl, 1.5 CaCl$_2$, 1 MgCl$_2$, 10 Hepes and 10 glucose (pH 7.4, 310 mOsm). HeLa cells were pre-incubated at 37°C with either 10 µM W123

(Cayman Chemical, Ann Arbor, MI, USA; catalog. no. 10010992) for 20 min, 10 μM W146 (Tocris, Bristol, UK; catalog. no. 3602) for 10 min, 100 nM JTE-013 (Tocris; catalog. no. 2392) for 10 min, 500 ng mL$^{-1}$ pertussis toxin (Tocris; catalog. no. 3097) overnight, 10 μM gallein (Tocris; catalog. no. 3090) for 30 min or 1 μM YM-254890 (Tocris; catalog. no. 7352) for 30 min. Then, 100 nM S1P (Sigma-Aldrich. St Louis, MO, USA; catalog. no. 73914) or 500 nM SEW-2871 (Tocris; catalog. no. 2284) in isotonic solution were applied to activate LRRC8/VRAC.

## FRET measurements

FRET experiments were performed essentially as described previously (Klüssendorf et al., 2024; König et al., 2019). Images were acquired using a high-speed setup of Leica Microsystems (Wetzlar, Germany) (Dmi6000B stage, 63×/1.4 oil objective, high-speed external Leica wheels with Leica FRET set filters (11522073), EL6000 light source, DFC360 FX camera, controlled by the Las AF software platform). All experiments were conducted at room temperature. Before imaging, the growth medium was removed and cells were washed three times with an isotonic solution. Sensitised emission FRET images were acquired in the donor, acceptor and FRET channels. Acquisition parameters remained the same for all the channels (8 × 8 binning, gain 1, 100 ms exposure time, illumination intensity 2). Corrected FRET (cFRET) values were calculated according to the following equation (Jiang & Sorkin, 2002):

$$cFRET = \frac{I^{DA} - I^{DD}\beta - I^{AA}\gamma}{I^{AA}}$$

where $I^{DD}$ is the emission intensity of the donor channel, $I^{AA}$ is the acceptor channel and $I^{DA}$ is the FRET channel. $\beta$ and $\gamma$ are the correction factors for the donor bleed-through and acceptor cross-excitation. The calculation of correction factors was described previously (König et al., 2019). cFRET maps were determined by hand-drawn regions of interest and were processed with the PixFRET plugin (Feige et al., 2005) (threshold set to 1, Gaussian blur to 2) with a self-written macro. cFRET values of individual cells were normalised to their mean cFRET in the isotonic solution. Hypotonic imaging buffer (250 mOsm) was the same as isotonic, but with only 105 mM NaCl. Hypertonic imaging buffer (500 mOsm) was as an isotonic buffer supplemented with 160 mM mannitol.

## Electrophysiology

Whole-cell voltage-clamp experiments were performed in isotonic extracellular solution. LRRC8/VRAC currents were elicited by perfusion of cells with isotonic solution containing 100 nM S1P or 500 nM SEW-2871. Subsequently, cells were perfused with hypotonic solution containing (in mM) 75 NaCl, 6 KCl, 1 MgCl$_2$, 1.5 CaCl$_2$, 10 glucose and 10 Hepes, pH 7.4, with NaOH (160 mOsm) to saturate the LRRC8/VRAC conductance. Pipette solution contained (in mM) 40 CsCl, 100 Cs-methanesulfonate, 1 MgCl$_2$, 1.9 CaCl$_2$, 5 EGTA, 4 Na$_2$ATP and 10 Hepes, pH 7.2, with CsOH (290 mOsm). Osmolarities of all solutions were assessed with an OM 807 freezing-point osmometer (Vogel MedTec, Fernwald, Germany). All experiments were performed at constant room temperature of 20–22°C. Currents were recorded with an EPC-10 USB patch clamp amplifier and PatchMaster software (HEKA Elektronik, Lambrecht, Germany) as described previously (Kashyap et al., 2024). Patch pipettes had a resistance of 3–5 MΩ. Currents were sampled at 5 kHz and low-pass filtered at 1 kHz. The holding potential was −30 mV. The standard protocol for measurement of the time course of LRRC8/VRAC current activation consisted of a 0.6 s step to −80 mV followed by a 2.6 s ramp from −100 to 100 mV, which was applied every 12 s. Readout for VRAC current was steady-state, whole-cell current at −80 mV normalised to cell capacitance subtracted by baseline current density at −80 mV before the application of stimulating solution. The voltage protocol, applied before the standard protocol and after 10 min of measurement, consisted of 2 s steps from −120 to 100 mV, with 20 mV increments, preceded and followed by a 0.5 s step to −80 mV every 5 s. The voltage-step protocol confirmed VRAC-typical properties of outward rectification and depolarization-dependent inactivation (König & Stauber, 2019) for S1P-induced currents. All experiments that included the external application of pharmacological modulators were performed only after stability of the baseline was reached (time-course control protocols lasting 1 min). At the end of some experiments, the VRAC inhibitors 4-[(2-butyl-6,7-dichloro-2-cyclopentyl-2,3-dihydro-1-oxo-1H-inden-5-yl)oxy]butanoic acid (DCPIB) (20 or 100 μM; Tocris; catalog. no. 1540) or carbenoxolone (100 μM, Tocris; catalog. no. 3096) were applied to confirm the nature of the currents.

## SDS-PAGE and immunoblotting

Cells were collected by scraping on ice in pre-cooled RIPA buffer (150 mM NaCl, 50 mM Tris, pH 8.0, 5 mM EDTA, pH 8.0, 1% NP-40, 0.5% sodium deoxycholate, 0.1% SDS) containing proteinase inhibitor cocktail (Roche, Basel, Switzerland) and phosphatase inhibitor cocktail (Sigma-Aldrich; catalog. no. P5726). The suspension was incubated on ice for 30 min and vortexed every 10 min. After subsequent centrifugation at 16 000 *g* at 4°C for 20 min, the supernatant was mixed with

SDS sample buffer. Cell lysates were separated by 10% SDS-PAGE and transferred to nitrocellulose membranes (Macherey-Nagel, Düren, Germany) at 400 mA for 90 min. After the transfer, the membranes were blocked with 5%-bovine serum albumin in Tris-buffered saline-Tween 20 (TBS-T) solution, containing 20 mM Tris, pH 7.6, 150 mM NaCl and 0.02% Tween-20 for 1 h at room temperature, and subsequently incubated with primary antibodies overnight at 4°C. Primary antibodies and dilutions used: PKD (Abcam; catalog. no. ab131460; RRID: AB_11157105): dilution 1:1000; phosphorylated PKD (p-PKD) (Cell Signaling Technology, Danvers, MA, USA; catalog. no. 2054; RRID:AB_2172539): dilution 1:1000, S1PR1 (Abcam; catalog. no. ab233386; RRID:AB_2928162): dilution 1:1000; GAPDH (Abcam; catalog. no. ab9485; RRID:AB_307275): dilution 1:5000. The primary antibodies against LRRC8A were kindly provided by Thomas J. Jentsch (Leibniz-Institut für Molekulare Pharmakologie), applied at the dilution 1:1000, and were described previously (Voss et al., 2014). After three washes for 10 min each with TBS-T, the membranes were incubated with horseradish peroxidase-conjugated secondary antibody (Abcam; catalog. no. ab6721; RRID:AB_955447), dilution 1:10,000 for 1 h at room temperature, washed with TBS-T and finally developed with enhanced chemiluminescence reagent (SuperSignal kit; Thermo Fisher Scientific) on a digital imaging system (Azure Biosystems, Dublin, CA, USA; #1.6.4.1229). Protein levels were quantified using Fiji software (https://fiji.sc/).

### mRNA expression analysis

To analyse expression of S1PR1-5 by reverse transcriptase-PCR (RT-PCR), HeLa cells were plated in a T25 flask at one-third of confluency. Then, 48 h later, cells were harvested by detaching with trypsin and centrifugation at 700 *g* for 3 min. Cell pellets were washed with PBS and stored at −20°C until further processing. Total RNA was isolated using NucleoSpin RNA (Macheray-Nagel) in accordance with the manufacturer's instructions, and 1 µg of total RNA was used for reverse transcription with a Hiscript III Q RT SuperMix for qPCR (+gDNA wiper) cDNA synthesis kit (Vazyme, Nanjing, China) in accordance with the manufacturer's instructions. Expression of the target mRNA was qualified using ChamQ Universal SYBR qPCR Master Mix (Vazyme) with following forward and reverse, respectively, primer pairs. For *S1PR1*: 5′-TTCCACCGACCCATGTACTAT-3′ and 5′-GCGAGGAGACTGAACACGG-3′; for *S1PR2*: 5′-CATCGTCATCCTCTGTTGCG-3′ and 5′-GCCTGC CAGTAGATCGGAG-3′; for *S1PR3*: 5′-CGGCATCGC

TTACAAGGTCAA-3′ and 5′-GCCACGAACATACTGC CCT-3′; for *S1PR4*: 5′-GACGCTGGGTCTACTATTG CC-3′ and 5′-CCTCCCGTAGGAACCACTG-3′; for *S1PR5*: 5′-GCGCACCTGTCCTGTACTC-3′ and 5′-GTT GGTGAGCGTGTAGATGATG-3′; for *GAPDH*: 5′-AC AGTCAGCCGCATCTTCTT-3′ and 5′-GTTAAAAGCA GCCCTGGTGA-3′. PCR amplification was performed with 40 cycles of alternating 95°C for 10 s and 60°C for 30 s after an initial denaturation at 95°C for 5 min. After amplification, samples were mixed with loading buffer (10× FastDigest Green Buffer; Thermo Fisher Scientific), separated parallel to a GeneRuler DNA Ladder Mix (Thermo Fisher Scientific) in 2% (w/v) agarose gels prepared with Tris-acetate-EDTA buffer and visualised with Roti-GelStain (Carl-Roth, Karlsruhe, Germany).

### Statistical analysis

Data parameters are expressed as the mean ± SD of *n* independent experiments. Statistical significance was determined using the Mann–Whitney *U* test in Prism (GraphPad Software Inc., San Diego, CA, USA). $P < 0.05$ was considered statistically significant.

## Results

### FRET-based assessment of LRRC8/VRAC activation by S1P

Previous results obtained in murine macrophage and microglial cells suggested that S1P activates anion currents similar to VRAC currents (Burow et al., 2015; Zahiri et al., 2021). We intended to track the S1P-induced iso-smotic LRRC8/VRAC activation using a non-invasive FRET optical sensor (König et al., 2019). To this end, we transfected HeLa cells with LRRC8A-Cerulean and LRRC8E-Venus and monitored FRET (Fig. 1*A*). We observed a significant decrease in the corrected FRET (cFRET) value, indicating LRRC8/VRAC activation, upon switching from isotonic (340 mOsm) to hypotonic (250 mOsm) solution; this could be reversed when cells were returned to isotonic bath solution (Fig. 1*B*).

We also observed a cFRET drop indicating channel activation in the presence of 10 nM S1P in isotonic bath solution (Fig. 1*C* and *F*), but not with its vehicle, methanol (Fig. 1*F*). cFRET returned to initial baseline values when the extracellular solution was replaced by an S1P-containing hypertonic solution (500 mOsm). Relative reduction of cFRET was less when induced by S1P compared to that induced by hypotonicity.

We next sought to identify which receptor subtype is responsible for S1P-induced LRRC8/VRAC activation. Pre-incubation of cells with 10 µM W123, an S1PR1

antagonist, reduced the effect of S1P (Fig. 1*D* and *F*). By contrast, the S1PR2 selective blocker JTE-013 (100 nM) had no effect on the S1P-induced cFRET reduction (Fig. 1*F*). Moreover, the application of an isotonic solution containing 500 nM of the selective S1PR1 agonist SEW-2871 (Sanna et al., 2004) was able to reversibly decrease cFRET similarly to 10 nM S1P (Fig. 1*E* and *F*). Taken together, these results suggest that S1P activates LRRC8/VRAC in HeLa cells through the S1PR1 receptor.

## S1PR1 determines activation of endogenous LRRC8/VRAC by S1P

We next tested S1P induction of VRAC currents by whole-cell patch clamp experiments in HeLa cells. Upon application of S1P at concentrations of 10 or 100 nM, which were previously used to activate LRRC8/VRAC (Burow et al., 2015; Chu et al., 2023; Furuya et al., 2021; Zahiri et al., 2021), we observed outwardly rectifying currents with voltage-dependent deactivation that are

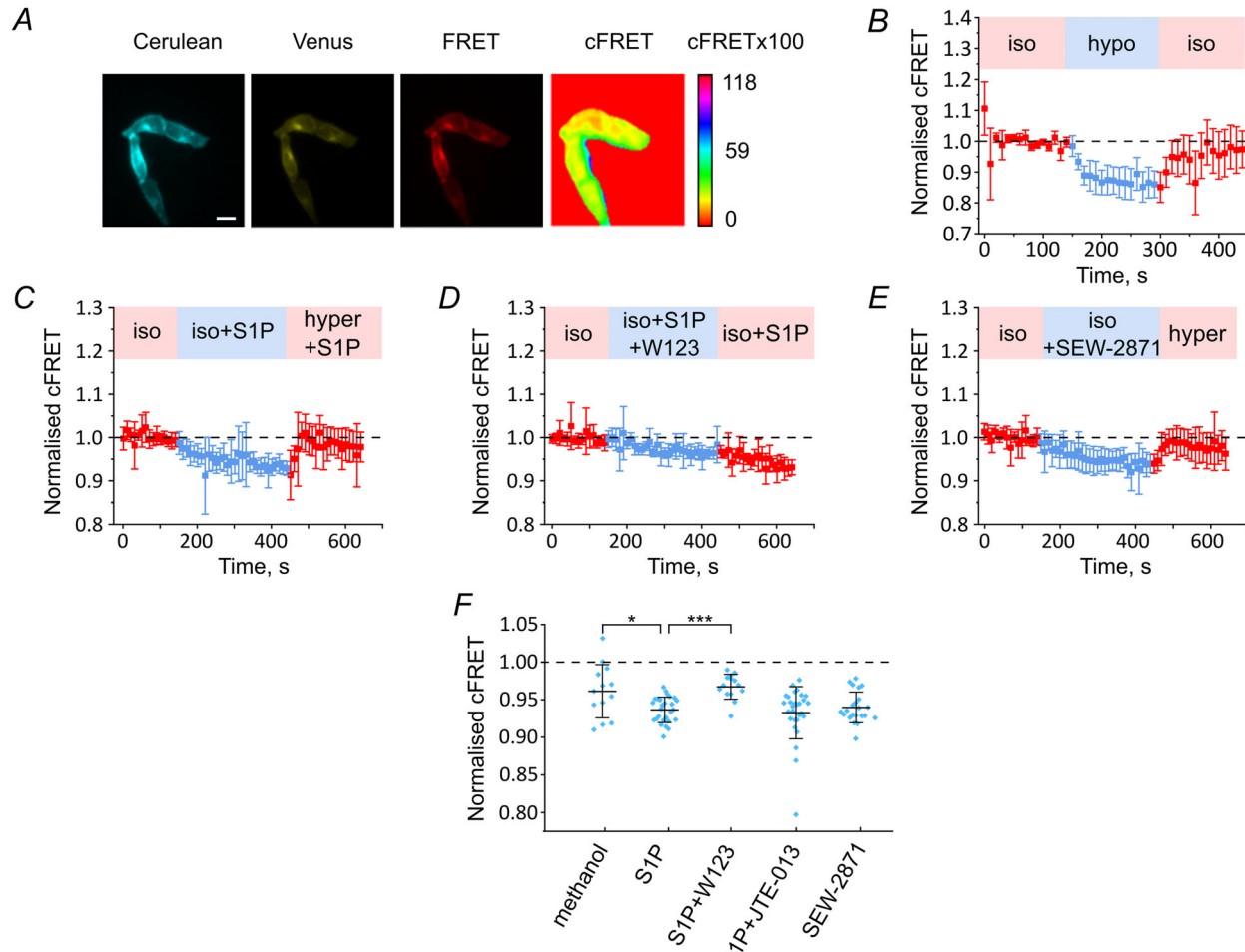

**Figure 1. cFRET changes reflect LRRC8/VRAC activation by S1P through S1PR1**
*A*, images of HeLa cells transfected with LRRC8A-Cerulean/LRRC8E-Venus. Showing the three channels needed for cFRET calculation: Cerulean (donor), Venus (acceptor) and FRET channels, as well as the cFRET map of the transfected cells calculated from the three channels using the pixFRET plugin of ImageJ. The calibration bar represents the cFRET values and their respective colour codes in look-up-table (LUT). Scale bar = 20 μm. *B*, cFRET normalised to the isotonic conditions during switching from isotonic to hypotonic solution (*n* = 7 cells). *C*, cFRET drop during buffer exchange to 10 nM S1P-containing isotonic solution (*n* = 27). *D*, normalised cFRET of the HeLa cells pre-incubated for 20 min with S1PR1 antagonist W123 at 10 μm during buffer exchange to 10 nM S1P and W123-containing isotonic buffer (*n* = 14). *E*, cFRET drop during buffer exchange to 500 nM SEW-2871-containing isotonic solution (*n* = 22). *F*, quantification of the normalised cFRET of HeLa cells challenged with isotonic supplemented with: S1P solvent methanol (*n* = 13), S1P (*n* = 27), as in (*B*), S1P and W123 (*n* = 14), as in (*C*), S1P and 100 nM JTE-013 (*n* = 29) and SEW-2871 (*n* = 22), as in (*D*). Data represent mean of last 10 points per condition of individual cells and the mean ± SD *$P$ = 0.0229, ***$P$ = 0.00000223 by the Mann–Whitney *U* test compared to S1P treatment, otherwise not significant ($P$ = 0.720 for S1P+JTE-013; $P$ = 0.643 for SEW-2871). [Colour figure can be viewed at wileyonlinelibrary.com]

typical for LRRC8/VRAC (Fig. 2A and C). Their inhibition by the VRAC blockers DCPIB and carbenoxolone suggested that the currents were indeed mediated by LRRC8/VRAC (Fig. 2A–C). Expectedly, the currents were absent in LRRC8A KO HeLa cells lacking the essential VRAC subunit (Yang et al., 2019) upon application of 100 nM S1P (Fig. 2C and D).

Pre-incubation of cells with the S1PR1 antagonists W123 or W146 (10 μM) abolished VRAC currents upon S1P application (Fig. 3A and B). However, we still observed VRAC activation in cells pre-treated with the S1PR2 antagonist JTE-013 (Fig. 3A and B). Application of the S1PR1 agonist SEW-2871 induced VRAC currents just like S1P (Fig. 3B). These findings confirmed our FRET results and, although RT-PCR showed the expression of all five S1PR1–S1PR5 paralogues in HeLa cells (Fig. 3C), further corroborate the idea that S1P activates LRRC8/VRAC via S1PR1.

Moreover, S1P failed to activate VRAC in a genomically S1PR1-depleted (S1PR1 KO) HeLa cell line (Fig. 4A and B). S1P-induced currents in S1PR1 KO HeLa cells could be restored by heterologous expression of either S1PR1 or, surprisingly, S1PR2 (Fig. 4A and C). This suggests that, although S1PR1 is required for S1P induction of LRRC8/VRAC currents at endogenous expression levels, overexpression of S1PR2 can compensate for the loss of S1PR1.

## S1P activates LRRC8/VRAC through G$\beta\gamma$ dimers

To explore the signalling pathway downstream of S1PR1, we tested the roles of heterotrimeric G proteins in S1P-induced LRRC8/VRAC activation. Because S1PR1 couples exclusively to heterotrimeric Gi proteins (Xiao et al., 2019), we first tested for the role of G$\alpha$i-G$\beta\gamma$ complexes. Pertussis toxin (PTX) inactivates the $\alpha$ subunit of Gi proteins through ADP ribosylation, thus preventing or slowing down its dissociation from the G$\beta\gamma$ subunit (Mangmool & Kurose, 2011). Pre-treatment of HeLa cells with 500 ng mL$^{-1}$ PTX overnight did not diminish S1P-induced LRRC8/VRAC currents (Fig. 5A and D),

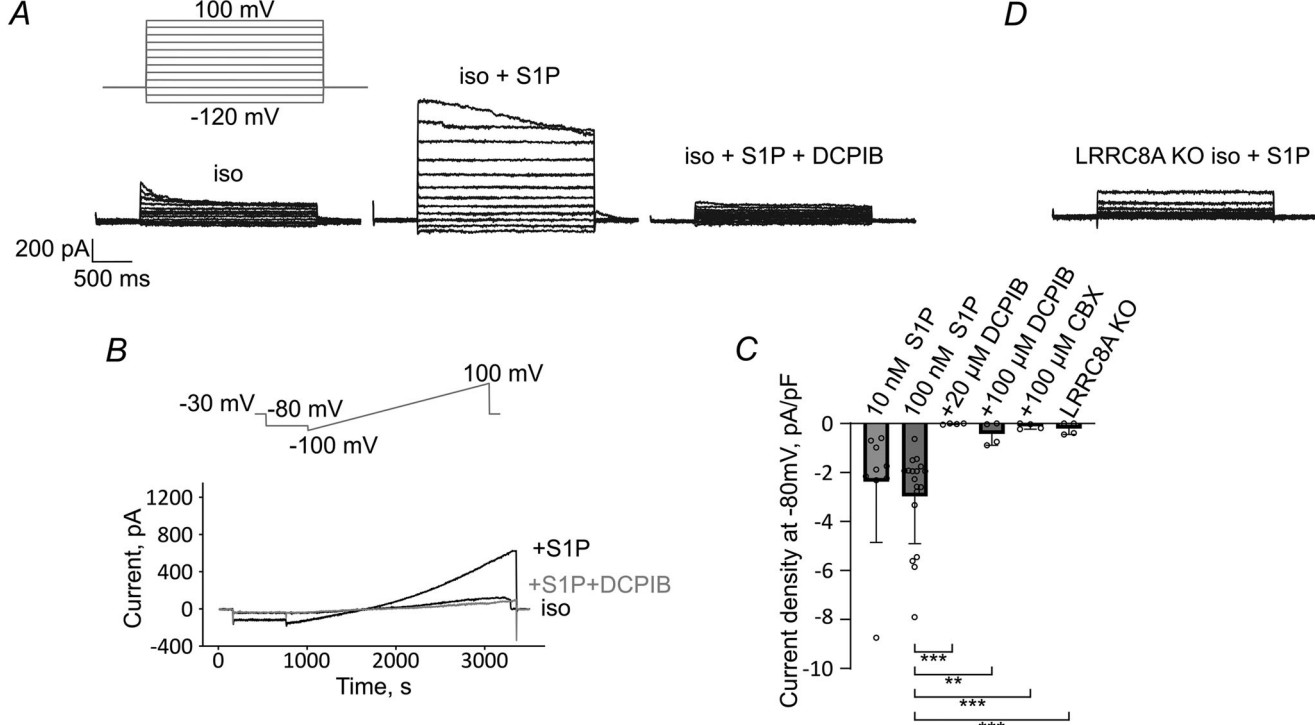

**Figure 2. S1P induces LRRC8-mediated currents**

*A*, representative current traces of LRRC8/VRAC activated by 100 nM S1P in a HeLa cell measured using the depicted protocol. Here, 20 μM DCPIB was applied to verify that currents are mediated by LRRC8/VRAC. *B*, time course of VRAC currents measured using a voltage-ramp protocol (top) in isotonic buffer and after application of 100 nM S1P and subsequently additionally DCPIB in a representative HeLa cell. *C*, quantification of current density at −80 mV. Values from individual cells (circles) and their mean ± SD are shown. Using the Mann–Whitney *U* test, the following conditions were compared to HeLa WT cells treated with 100 nM S1P ($n = 18$): HeLa WT cells treated with 10 nM S1P ($n = 9$, not significant, $P = 0.194$) or with 100 nM S1P and additionally 20 μM DCPIB ($n = 4$, ***$P = 0.000273$), 100 μM DCPIB ($n = 4$, **$P = 0.00109$) or 100 μM carbenoxolone (CBX, $n = 4$, ***$P = 0.000273$), as well as HeLa LRRC8A KO cells after 100 nM S1P application ($n = 4$, ***$P = 0.000273$). *D*, representative current traces of LRRC8A KO HeLa cells after application of 100 nM S1P measured using the depicted protocol in (*A*).

excluding the role of the PTX-sensitive Gi. Next, we assessed the role of G$\beta\gamma$ complexes in LRRC8/VRAC activation by performing measurements in HeLa cells treated with gallein, a G$\beta\gamma$ inhibitor (Lehmann et al., 2008). Gallein binds to the effector binding sites of G$\beta$, blocking the interaction of G$\beta\gamma$ with effectors (Siripurapu et al., 2017). Incubation of HeLa cells for 30 min with gallein prevented S1P-induced LRRC8/VRAC activation, pointing towards G$\beta\gamma$ as a transducer of S1PR1-mediated LRRC8/VRAC activation (Fig. 5*B* and *D*).

PTX-insensitive heteromeric G protein signalling is mainly mediated by G$\alpha$q-dependent activation of PLC$\beta$ or by pathways involving the Rho monomeric proteins and PLC$\varepsilon$ (Citro et al., 2007; Singer et al., 1997). Moreover, it was reported that Gq hierarchically controls Gi-Ca$^{2+}$ signalling (Pfeil et al., 2020). We therefore decided to check for a role of Gq signalling in LRRC8/VRAC activation. To this end, we applied YM-254890, which prevents the guanine nucleotide exchange on G$\alpha$q, keeping it in its inactive G$\alpha$q-GDP form (Mizuno & Itoh, 2009), and also inhibits Gs (Peng et al., 2021). Pre-incubation of HeLa cells with 1 µM YM-254890 did not affect LRRC8/VRAC activation by S1P (Fig. 5*C* and *D*). These data suggest that

the Gq and Gs families of G proteins are dispensable for LRRC8/VRAC activation.

### S1P causes PKD phosphorylation

G$\beta\gamma$ dimers from heterotrimeric G$\alpha$i-G$\beta\gamma$ proteins are known to specifically stimulate PLC$\beta$2 and PLC$\beta$3 (Falzone & MacKinnon, 2023; Pfeil et al., 2020; Rebres et al., 2011), leading to the activation of a variety of protein kinases, including protein kinase C (PKC) and PKD. Because PKD was previously suggested to activate LRRC8/VRAC (König et al., 2019), we assessed PKD phosphorylation upon hypotonic and S1P stimuli. Incubation of HeLa cells in hypotonic buffer caused an increase in the ratio of p-PKD to total PKD (Fig. 6*A* and *B*). We observed a similar effect when cells were incubated in 100 nM S1P (Fig. 6*C* and *D*). Strikingly, pre-incubation with gallein or W123 diminished the time-dependent increase in p-PKD protein levels (Fig. 6*E* and *F*). These results emphasize the role of S1PR1 and G$\beta\gamma$ dimers in mediating the S1P-induced cellular response.

### Heterologous expression of G$\beta\gamma$-sensitive PLC$\beta$3 facilitates S1P-mediated LRRC8/VRAC activation in HEK293 cells

To further challenge the notion that LRRC8/VRAC is activated by S1P through G$\beta\gamma$-PLC$\beta$ signalling, we tested S1P induction of VRAC currents in HEK293 cells, which were reported to express less of the G$\beta\gamma$-sensitive PLC$\beta$ isoforms PLC$\beta$2 and PLC$\beta$3 compared to HeLa cells (Lau et al., 2013). Indeed, we observed almost no detectable LRRC8/VRAC activation upon S1P application in HEK293 cells (Fig. 7*A*, *B* and *E*), although hypotonicity could activate VRAC currents in both HeLa and HEK293 cells (Fig. 7*A*, *B* and *F*). Heterologous expression of PLC$\beta$3, which is recruited to and oriented in the membrane by G$\beta\gamma$ (Falzone & MacKinnon, 2023), restored S1P-induced VRAC currents in wild-type HEK293 cells (Fig. 7*C* and *E*), but not in LRRC8 KO HEK293 cells that lack all five LRRC8 paralogues and do not produce hypotonicity-induced VRAC currents (Fig. 7*D*–*F*). These data support the crucial role of G$\beta\gamma$-PLC$\beta$ interaction in the conduction of S1P-induced signalling to LRRC8/VRAC channel activation.

### Discussion

The involvement of G proteins in LRRC8/VRAC activation and regulation has been implicated in numerous studies (Estevez et al., 2001; Voets et al., 1998). The physiological relevance of G protein-mediated isosmotic LRRC8/VRAC activation is of particular

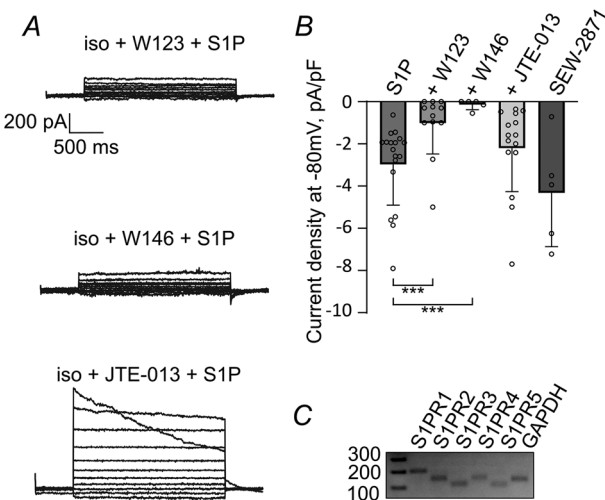

**Figure 3. Pharmacological inhibition of S1PR1 prevents S1P-induced LRRC8 currents**

*A*, S1P failed to induce LRRC8/VRAC current in cells pre-incubated with 10 µM W123 for 20 min (top) or 10 µM W146 for 10 min (middle), but S1P induced LRRC8/VRAC currents in cells pre-incubated with 100 nM JTE-013 for 10 min (bottom). *B*, quantification of current density at −80 mV. Values from individual cells (circles) and their mean ± SD are shown. Using the Mann–Whitney *U* test, the following conditions were compared to HeLa WT treated with 100 nM S1P (*n* = 18, values as in Fig. 2*C*): HeLa WT pre-incubated with 10 µM W123 (*n* = 12, **P = 0.000519), 10 µM W146 (*n* = 5, ***P = 0.0000594) or 100 nM JTE-013 (*n* = 15, *P* = 0.0794) and HeLa WT after 500 nM SEW-2871 application in the absence of S1P (*n* = 5, *P* = 0.199). *C*, gel electrophoresis of RT-PCR products demonstrated mRNA expression of all five S1PR paralogues in HeLa cells.

interest because most mammalian cell types hardly experience extracellular hypoosmolarity under physiological conditions. Intracellular application of the non-hydrolysable GTP analogue GTPγS, which turns G proteins 'on', induced VRAC currents in various cell lines (Doroshenko et al., 1991; Doroshenko & Neher, 1992; Estevez et al., 2001; Voets et al., 1998). By contrast, the GDP analogue GDPβS, known to inhibit G proteins, prevented these currents (Burow et al., 2015; Voets et al., 1998). In astrocytes, stimulation of purinergic G protein-coupled receptors, P2YRs, also leads to an isovolumic LRRC8/VRAC activation (Hyzinski-García et al., 2014; Mongin & Kimelberg, 2005). There is a growing body of evidence that S1P is capable of activating LRRC8/VRACs under isovolumetric conditions in a variety of cell types (Burow et al., 2015; Chu et al., 2023; Furuya et al., 2021; Zahiri et al., 2021). S1P-mediated signalling has been shown to be involved in many physiological and pathophysiological processes such as cancer, diabetes and osteoporosis (Maceyka et al., 2012). It not only influences the complex reactions of the innate immune system, but also is of importance during the aberrant production of inflammatory cytokines in auto-inflammatory disorders and sepsis (Burow et al., 2015; Chi, 2011). Most of the known actions of S1P are mediated

by a family of five specific G protein-coupled receptors, S1PR1–S1PR5 (Hisano et al., 2012; Rosen et al., 2013).

Given that the signal transduction from S1PRs to LRRC8/VRAC has remained unclear, the present study investigates the individual components of this process. We made use of two complementary experimental approaches: on one hand, utilizing a non-invasive FRET sensor leaves intracellular signalling pathways unaffected; on the other hand, the whole-cell patch clamp allows monitoring endogenous currents and is the most sensitive technique, being especially applicable for the evaluation of small currents. Indeed, we found that, in HeLa cells, the amplitudes of the currents evoked by S1P were in the pA range (Fig. 2*A* and *B*), as previously reported for other cell types (Burow et al., 2015; Chu et al., 2023; Furuya et al., 2021; Zahiri et al., 2021). The currents were sensitive to DCPIB and carbenoxolone and augmented by the application of the hypotonic solution, confirming that the S1P-activated channel is indeed LRRC8/VRAC. As expected, both S1P-induced and hypotonicity-induced (Yang et al., 2019) currents were absent in LRRC8A-deficient HeLa cells. Furthermore, we discovered that the S1PR1 receptor antagonists W123 and W146 (Figs 1 and 3), as well as genomic depletion of S1PR1 in HeLa cells (Fig. 4), prevented

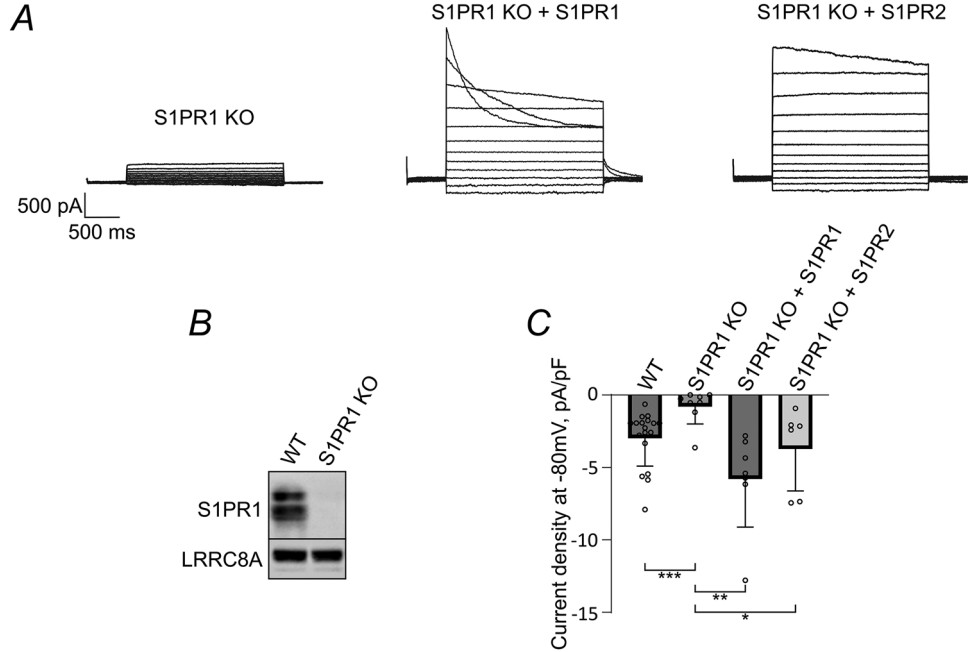

**Figure 4. Ectopic expression of S1PR1 or S1PR2 restores S1P-induced LRRC8 currents in S1PR1-deficient cells**

*A*, S1P failed to induce VRAC current in S1PR1 KO cell (left), but did induce currents in S1PR1 KO cells upon heterologous expression of S1PR1 (middle) or S1PR2 (right). *B*, immunoblotting against S1PR1 confirmed the knockout in S1PR1 KO HeLa cells. LRRC8A was used as loading control. *C*, quantification of current density at −80 mV after the application of 100 nM S1P. Values from individual cells (circles) and their mean ± SD are shown. Using the Mann–Whitney *U* test, the following conditions were compared to HeLa S1PR1 KO (*n* = 8): HeLa WT (values as in Fig. 2*C*, *n* = 18, \*\*\**P* = 0.000766) and HeLa S1PR1 KO after transient transfection of S1PR1 (*n* = 7, \*\**P* = 0.00124) or S1PR2 (*n* = 6, \**P* = 0.0123).

channel activation upon S1P ligand binding, pointing to a key involvement of S1PR1. This notion is further supported by our observation that S1PR2 inhibition with JTE-013 did not affect LRRC8/VRAC activation in HeLa cells (Figs 1 and 3). These results are consistent with a previous report on macrophages, where W123, but not JTE-013, abolished S1P-induced VRAC currents (Burow et al., 2015). However, overexpression of S1PR2 in S1PR1-deficient cells restored S1P-induced currents (Fig. 4), suggesting that, depending on expression levels, S1PR2 may play a role in S1P-induced VRAC activation. This aligns with a recent study on microglia, in which pharmacological blocking of either S1PR1 or S1PR2 abolished VRAC currents, indicating the requirement of both receptors in those cells (Chu et al., 2023). Although we detected mRNA for all five S1PRs, our results demonstrate that S1P-induced LRRC8 activation is predominantly mediated by S1PR1 in HeLa cells.

Endothelial cells express S1PR1, S1PR2 and S1PR3 (Lee et al., 1999), whereas S1PR4 is predominantly expressed in lymphatic and hematopoietic tissues and S1PR5 in the central nervous system (Takuwa et al., 2012). Whereas S1PR1 couples exclusively to heterotrimeric

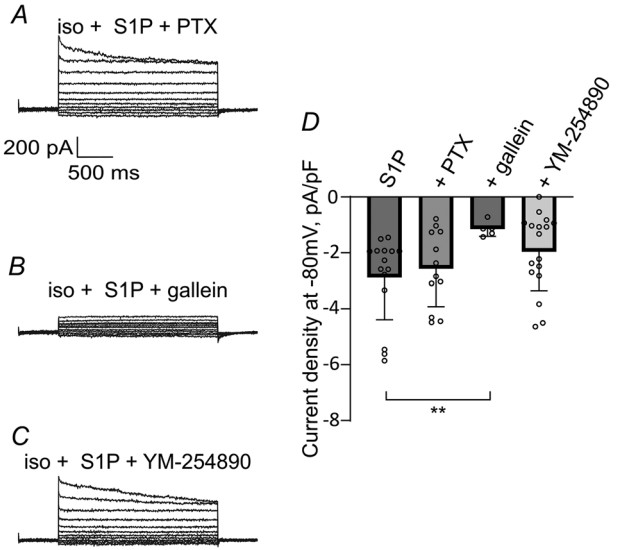

**Figure 5. The role of heterotrimeric G proteins in S1P-induced LRRC8/VRAC activation**

*A*, current traces of LRRC8/VRAC activated by 100 nM S1P in a representative HeLa WT cell pre-incubated with 500 ng mL$^{-1}$ PTX. *B*, lack of S1P induced VRAC currents in a cell pre-incubated with 10 μM gallein. *C*, S1P induced VRAC current in a cell pre-incubated with 1 μM YM-254890 for 30 min. *D*, quantification of current density at −80 mV. Values from individual cells (circles) and their mean ± SD are shown. The Mann–Whitney *U* test was employed for statistical analysis. After 100 nM S1P application, the following conditions were compared to HeLa WT without inhibitors (S1P, values as in Fig. 2*C*, *n* = 18): HeLa WT pre-incubated with 500 ng mL$^{-1}$ PTX (*n* = 12, *P* = 0.723), 10 μM gallein (*n* = 5, \*\**P* = 0.00113), 1 μM YM-254890 (*n* = 17, *P* = 0.103).

Gi proteins, S1PR2 and S1PR3 exert overlapping yet distinct intracellular signalling (Rosen et al., 2013). We ruled out the participation of Gq and PTX-sensitive Gi signalling in S1P-induced LRRC8/VRAC activation by use of specific blockers (Fig. 5). Gq is known to activate membrane-bound PLC, leading to inositol trisphosphate (IP$_3$)-mediated elevation of intracellular Ca$^{2+}$. Notably, the lack of an increase in intracellular Ca$^{2+}$ levels upon application of S1P onto macrophages, suggesting insensitivity of Gq to S1P, also argues against an involvement of Gq signalling in S1P-induced VRAC activation (Burow et al., 2015).

Although Gα subunits are often portrayed as the main players in GPCR signalling, Gβγ subunits, especially of Gi heterotrimers, are also capable of signalling to downstream effectors (Falzone & MacKinnon, 2023; Kadamur & Ross, 2013; Pfeil et al., 2020; Smrcka, 2008). Gi-Gβγ was reported to mediate the activation of G protein-gated, inwardly rectifying potassium (GIRK) and Kv7/KCNQ channels (Jin et al., 2002; Stott et al., 2015), as well as to recruit and activate membrane PLCβ (Falzone & MacKinnon, 2023; Kadamur & Ross, 2013). Strikingly, blocking Gβγ effector interactions with gallein suppressed S1P-induced LRRC8/VRAC activation (Fig. 5), highlighting the role of Gi-Gβγ in this process. Given the insensitivity of the currents to PTX and YM-254890, we suggest that Gβγ activating VRAC might originate from the PTX-insensitive Gi protein family. PTX ADP-ribosylates members of the Gαi subfamily at a particular cysteine residue, rendering them incapable of coupling to GPCRs. The only Gi protein family member that lacks this cysteine and therefore sensitivity to PTX is Gαz (Keen et al., 2022). Remarkably, a recent study showed that, among the Gi protein family, S1PR1 couples predominantly to PTX-insensitive Gαz (Ono et al., 2023). It is therefore tempting to speculate that S1P binding to S1PR1 causes the dissociation of the Gαz-Gβγ heterotrimer and consequent Gβγ-mediated LRRC8/VRAC activation. Gβγ activates GIRK or Kv7/KCNQ channels by direct interaction (Jin et al., 2002; Stott et al., 2015), whereas the involvement of diacylglycerol signalling (König et al., 2019) suggests further signal transduction for LRRC8/VRAC activation, possibly via Gβγ-sensitive isoforms of PLCβ. Indeed, we could not detect S1P-induced LRRC8 currents in HEK293 cells (Fig. 7) that were reported to express less Gβγ-sensitive PLCβ2/3 than HeLa cells (Lau et al., 2013). When transfected with PLCβ3, however, HEK293 cells exhibited LRRC8/VRAC activation upon S1P application, similar to HeLa cells (Fig. 7). The evidence of Gαz-Gβγ involvement suggests a particular physiological importance in pancreatic β cells and throughout the central nervous system where Gαz is highly expressed (Casey et al., 1990; Kimple et al., 2005). Further studies are needed to clarify the role of

Gβγ-sensitive isoforms of PLCβ upstream of PKD and LRRC8/VRAC activation in various cell types.

Considering the broad spectrum of reported compounds that can activate LRRC8/VRAC (Kolobkova et al., 2021), it is important to clarify where the distinct signalling pathways resulting in channel activation merge. PLCβ cleaves phosphatidylinositol-4,5-bisphosphate (PIP$_2$) into IP$_3$ and diacylglycerol, which activates PKC and PKD. The diacylglycerol analogue

phorbol-12-myristate-13-acetate (PMA) activates LRRC8 under isotonic conditions (König et al., 2019). Several studies also reported the involvement of PKC or PKD in VRAC activation and modulation (Hermoso et al., 2004; König et al., 2019; Rudkouskaya et al., 2008). Notably, pharmacological inhibition of PKD, but not PKC, was reported to impair hypotonicity-induced VRAC activation in HEK293 cells (König et al., 2019). PKD activation requires the phosphorylation of two

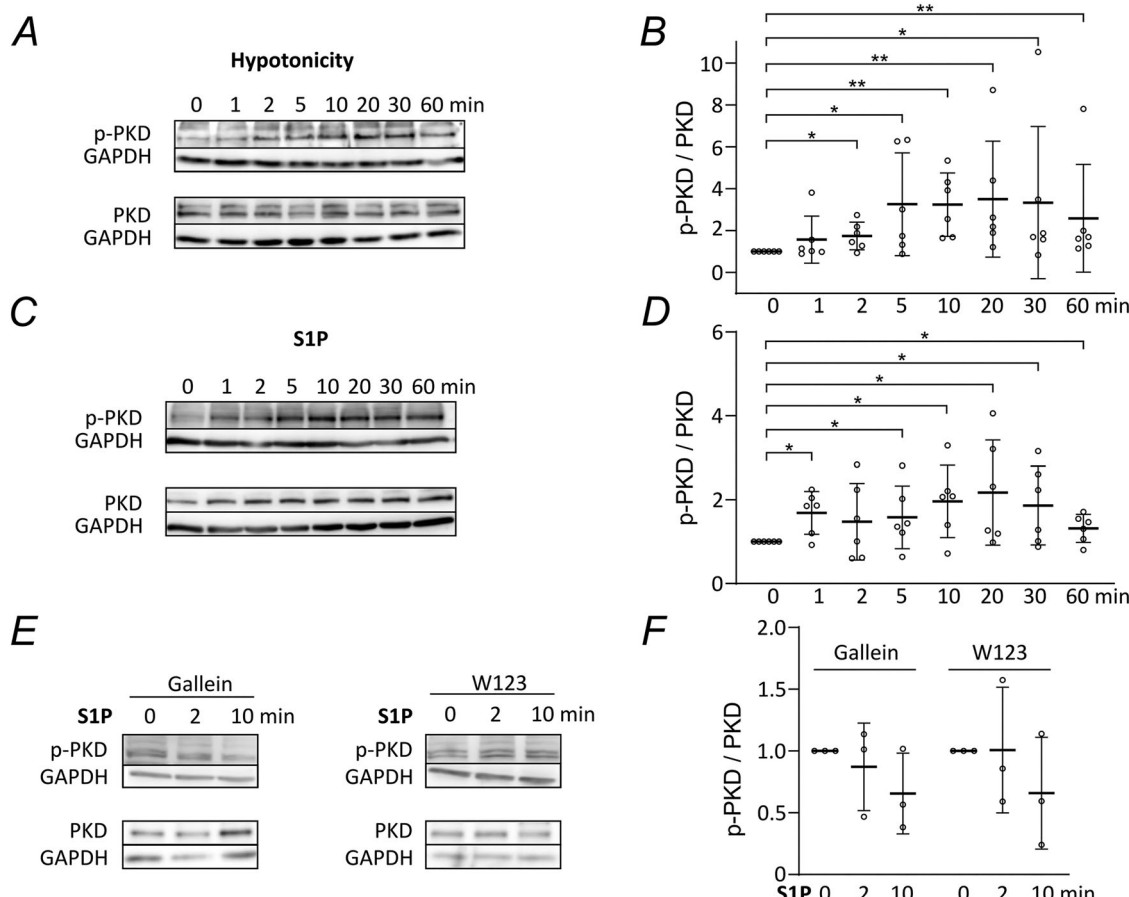

**Figure 6. S1P induces phosphorylation of PKD via Gβγ signalling**

*A*, representative immunoblots with antibodies against phosphorylated (p-PKD, Ser744/748) and total PKD on HeLa cell lysates collected after indicated times in 50% hypotonic solution. *B*, signal intensities of immunoblots as in (*A*) were quantified and normalised to GAPDH signals. Data represent values from individual experiments (circles) and the mean ± SD of the p-PKD/PKD ratio normalised to *t* = 0 min for six independent experiments. The Mann–Whitney *U* test was employed for statistical analysis compared to 0 min: 1 min: *P* = 0.364; 2 min: **P* = 0.0476; 5 min: **P* = 0.0476; 10 min: ***P* = 0.00217; 20 min: ***P* = 0.00217; 30 min: **P* = 0.0476; 60 min: ***P* = 0.00217. *C*, lysates of HeLa cells treated with isotonic solution containing 100 nM S1P for indicated times were immunoblotted against p-PKD (Ser744/748) and total PKD. Representative immunoblots for six independent experiments are shown. *D*, signal intensities from blots as in (*C*) normalised to GAPDH were used to calculate the p-PKD/PKD ratio. Signals are shown as the mean ± SD for six independent experiments (circles). The Mann–Whitney *U* test was employed for statistical analysis compared to 0 min: 1 min: **P* = 0.0476; 2 min: *P* = 0.364; 5 min: **P* = 0.0476; 10 min: **P* = 0.0476; 20 min: **P* = 0.0476; 30 min: **P* = 0.0476; 60 min: **P* = 0.0476. *E*, after pre-treatment with gallein for 30 min or W123 for 20 min, HeLa cells were incubated in isotonic solution containing 100 nM S1P for 0, 2 and 10 min. Cell lysates were immunoblotted with anti-p-PKD (Ser744/748) and anti-PKD antibodies. *F*, p-PKD/PKD ratios of signal intensities normalised to GAPDH levels shown as the mean ± SD for three independent experiments (circles). The Mann–Whitney *U* test was employed for statistical analysis compared to 0 min for gallein (2 min: *P* = 0.700; 10 min: *P* = 0.700) and W123 (2 min: *P* = 0.700; 10 min: *P* = 0.700).

serine residues in the kinase domain (Iglesias et al., 1998). Here, we demonstrate that both hypotonicity and S1P stimulation caused the phosphorylation of PKD in HeLa cells (Fig. 6). Although phosphorylation is still debated as being crucial for VRAC activation (Bertelli et al., 2021), activated PKDs could potentially directly phosphorylate plasma membrane-localised LRRC8/VRAC. LRRC8 subunits were detected in phospho-proteome screens with phosphorylation sites similarly located in their first intracellular loops and a PKD motif in the leucine-rich repeat domain (Olsen et al., 2006). Additionally, the involvement of $G\beta\gamma$ can be reconciled with PKD-mediated LRRC8/VRAC activation because gallein impaired S1P-induced PKD

phosphorylation (Fig. 6). With W123, we observed the same effect, again emphasizing the crucial role of S1PR1. Our results thus suggest that PKD is a point of convergence and integration of various signalling pathways leading to LRRC8/VRAC activation.

Our FRET measurements revealed conformational rearrangements of the C-terminal leucine-rich repeat domain upon S1P application, which may lead to pore opening for Cl⁻ and osmolyte transport (Deneka et al., 2021; Gaitán-Peñas et al., 2016; König et al., 2019). For LRRC8/VRACs, osmolyte transport has been shown to be induced by the same signals that activate Cl⁻ currents: hypotonic swelling activates both LRRC8-mediated currents and osmolyte transport (Qiu et al., 2014; Voss

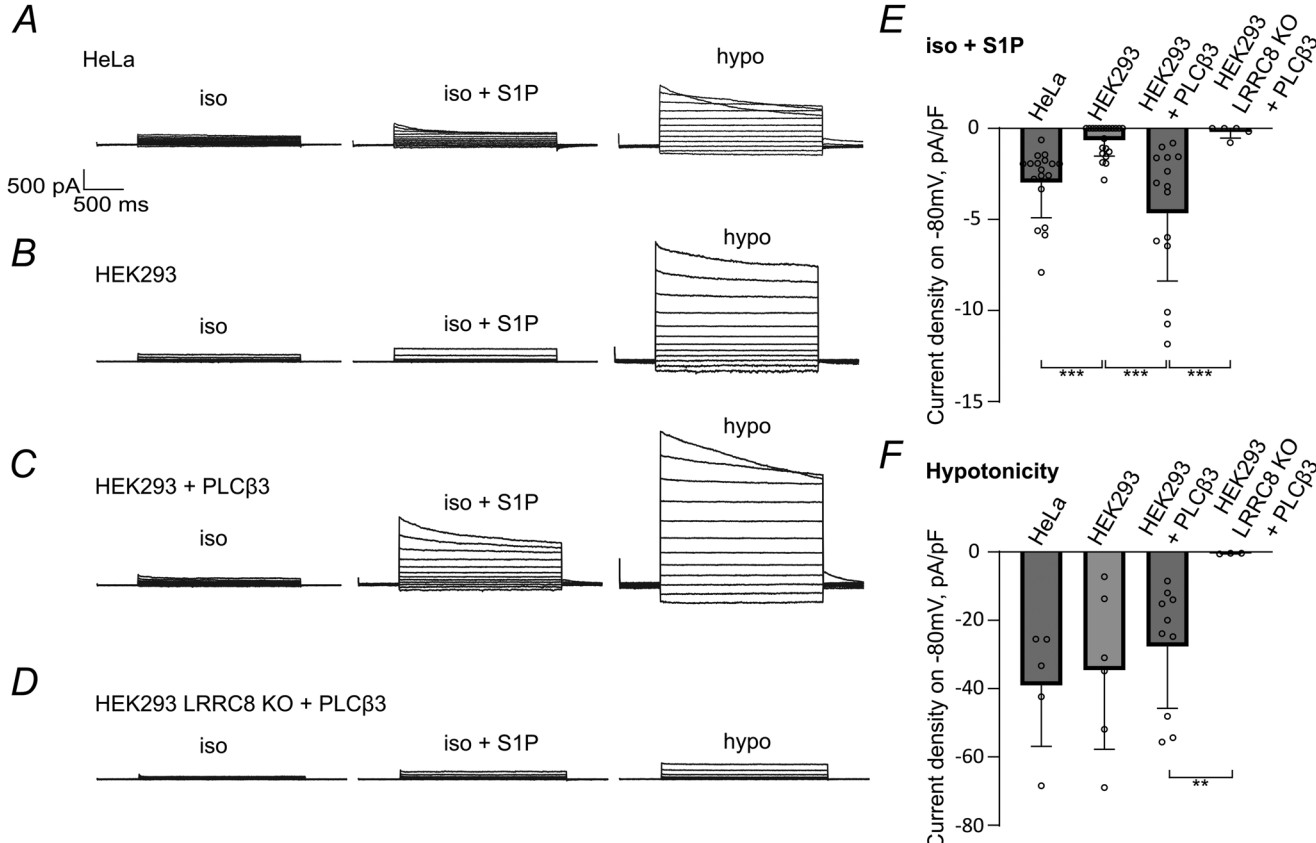

**Figure 7. Overexpression of PLCβ3 in HEK293 cells allows for S1P-induced LRRC8/VRAC activation**
*A*, representative current traces of LRRC8/VRAC activated by 100 nM S1P and subsequently by hypotonic solution aimed to saturate the LRRC8/VRAC conductance in a HeLa cell. *B*, representative current traces from a HEK293 cell showing that application of 100 nM S1P failed to induce VRAC current whereas hypotonicity-induced LRRC8/VRAC activation was intact. *C*, when overexpressing PLCβ3, HEK293 cell exhibited LRRC8/VRAC activation upon application of 100 nM S1P. *D*, neither S1P, nor hypotonicity induced LRRC8/VRAC currents upon overexpression of PLCβ3 in LRRC8 KO HEK293 cells. *E*, quantification of S1P-induced current densities at −80 mV. Values from individual cells (circles) and their mean ± SD are shown. Untransfected HEK293 WT ($n = 21$, ***$P = 0.00000372$) and LRRC8 KO transfected with PLCβ3 cDNA ($n = 5$, ***$P = 0.000129$) were compared to HEK293 WT transfected with PLCβ3 cDNA ($n = 15$) using the Mann–Whitney $U$ test. Values for HeLa are the same as in Fig. 2*C* ($n = 18$, ***$P = 0.000000459$ compared to HEK293 WT). *F*, quantification of hypotonicity-induced current densities at −80 mV. Values from individual cells (circles) and their mean ± SD are shown. Untransfected HEK293 WT ($n = 6$, $P = 0.713$) and LRRC8 KO transfected with PLCβ3 cDNA ($n = 3$, ***$P = 0.00699$) were compared to HEK293 WT transfected with PLCβ3 cDNA ($n = 10$). $P = 0.931$ between HEK293 WT and HeLa ($n = 5$) using the Mann–Whitney $U$ test.

et al., 2014), including ATP release (Gaitán-Peñas et al., 2016). Similarly, S1P induces both currents and ATP release (Burow et al., 2015; Chu et al., 2023; Furuya et al., 2021; Zahiri et al., 2021). Interestingly, a recent study showed that osmolyte transport by connexin hemichannels, which, just like LRRC8/VRAC, belong to large-pore channels, can be uncoupled from their ion channel function (Gaete et al., 2024).

To understand the complex molecular and cellular mechanisms of LRRC8/VRAC activation, it is necessary to elucidate how the respective cellular stimuli act on specific LRRC8 protein segments. From a structural point of view, the intracellular loops and leucine-rich repeat domains are of critical importance for channel activation (Kasuya & Nureki, 2022; König & Stauber, 2019; Sawicka & Dutzler, 2022; Yamada & Strange, 2018). The main mechanisms proposed to modulate VRAC activity include low intra-cellular ionic strength, oxidation, phosphorylation and G protein signalling. An oxidative milieu and reduced cytosolic ionic strength as a result of water influx in a hypotonic environment may act directly on the pore-forming proteins and the evoked conformational changes can contribute to channel opening (Bertelli et al., 2021, 2022; Syeda et al., 2016). Multiple studies implicate phosphorylation events as an obligatory part of swelling-induced VRAC activation (Bertelli et al., 2021). The present study suggests that phosphorylation plays a pivotal role not only in hypotonicity-, but also in iso-smotic, GPCR-mediated VRAC activation. It remains to be determined whether other LRRC8/VRAC stimuli partially share the described activating pathway, whether there are cell type-specific differences possibly as a result of variable LRRC8 composition, and, lastly, whether the target of phosphorylation is LRRC8/VRAC itself or another regulatory protein.

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

## Additional information

### Data availability statement

Source data and materials are available from the authors upon reasonable request.

### Competing interests

The authors declare that they have no competing interests.

### Author contributions

Y.K., S.P. and T.S. designed the research. Y.K. performed electrophysiological experiments and analysed data. S.P. performed FRET experiments and analysed data. Y.K., S.P. and A.K. performed immunoblotting. M.K. performed RT-PCR and analysed data. Y.K. drafted the manuscript. Y.K. and T.S. prepared the final version of the paper. All authors have read and approved the final version of the manuscript submitted for publication and agree to be accountable for all aspects of the work in ensuring that questions related to the accuracy or integrity of any part of the work are appropriately investigated and resolved. All persons designated as authors qualify for authorship, and all those who qualify for authorship are listed.

### Funding

The research was in part supported by the German Federal Ministry of Education and Research (BMBF, grant no. 031A314 to T.S.) and by the University of Punjab, Pakistan (S.P.). Open Access Funding was provided by the MSH Medical School Hamburg.

### Acknowledgements

We thank Tianbao Liu for initial experiments on PKD activation, Thomas Jentsch for the primary antibodies against LRRC8A and the LRRC8 KO HEK293 cells, Zhaozhu Qiu for the LRRC8A KO HeLa cells, Bryan Roth for S1PR plasmids, Andrei Kostritskii for critically reading the manuscript, and all members of the Stauber laboratory for helpful discussions.

### Keywords

GPCR signalling, LRRC8, sphingosine-1-phosphate, VRAC

## Supporting information

Additional supporting information can be found online in the Supporting Information section at the end of the HTML view of the article. Supporting information files available:

**Peer Review History**

