## [Peer Review History · The Journal of Physiology]

sphingosine-1-phosphate activates LRRC8 volume-regulated anion channels through G β γ signalling

Yulia Kostritskaia, Sumaira Pervaiz, Anna Klemmer, Malte Klüssendorf, and Tobias Stauber

DOI: 10.1113/JP286665

Corresponding author(s): Tobias Stauber (Tobias.Stauber@fu-berlin.de)

The following individual(s) involved in review of this submission have agreed to reveal their identity: Axel R. Conception (Referee #2)

Review Timeline:

Submission Date:	02-Apr-2024
Editorial Decision:	24-Apr-2024
Revision Received:	30-Sep-2024
Accepted:	15-Oct-2024

Senior Editor: Peking Fong

Reviewing Editor: Jorge Contreras

Transaction Report:

Dear Dr Stauber,

Re: JP-RP-2024-286665 "Sphingosine-1-phosphate activates LRRc8 volume-regulated anion channels through G β y signalling" by Yulia Kostrikskaia, Sumaira Pervaiz, Anna Klemmer, and Tobias Stauber

Thank you for submitting your manuscript to The Journal of Physiology. It has been assessed by a Reviewing Editor and by 2 expert referees and we are pleased to tell you that it is potentially acceptable for publication following satisfactory major revision.

Please address all the points raised and incorporate all requested revisions or explain in your Response to Referees why a change has not been made. We hope you will find the comments helpful and that you will be able to return your revised manuscript within 3 months. If you require longer than this, please contact journal staff: jp@physoc.org. Please note that this letter does not constitute a guarantee for acceptance of your revised manuscript.

REVISION CHECKLIST:

We look forward to receiving your revised submission.

Yours sincerely,

Peying Fong
Senior Editor
The Journal of Physiology

REQUIRED ITEMS

- Author photo and profile. First or joint first authors are asked to provide a short biography (no more than 100 words for one author or 150 words in total for joint first authors) and a portrait photograph. These should be uploaded and clearly labelled together in a Word document with the revised version of the manuscript. See Information for Authors for further details.

- The Journal of Physiology funds authors of provisionally accepted papers to use the premium BioRender site to create high resolution schematic figures. Follow this link and enter your details and the manuscript number to create and download figures. Upload these as the figure files for your revised submission. If you choose not to take up this offer, we require figures to be of similar quality and resolution. If you are opting out of this service to authors, state this in the Comments section on the Detailed Information page of the submission form. The link provided should only be used for the purposes of this submission. Authors will be charged for figures created on this premium BioRender account if they are not related to this manuscript submission.

- Please upload separate high-quality figure files via the submission form.

- You must upload original, uncropped western blot/gel images (including controls) if they are not included in the manuscript. This is to confirm that no inappropriate, unethical or misleading image manipulation has occurred. These should be uploaded as 'Supporting information for review process only'. Please label/highlight the original gels so that we can clearly see which sections/lanes have been used in the manuscript figures. For more information, see: <https://physoc.onlinelibrary.wiley.com/hub/journal-policies#imagmanip>.

- Please ensure that the Article File you upload is a Word file.

- Papers must comply with the Statistics Policy: https://jp.msubmit.net/cgi-bin/main.plex?form_type=display_requirements#statistics.

In summary:

- If n {less than or equal to} 30, all data points must be plotted in the figure in a way that reveals their range and distribution. A bar graph with data points overlaid, a box and whisker plot or a violin plot (preferably with data points included) are acceptable formats.

- If $n > 30$, then the entire raw dataset must be made available either as supporting information, or hosted on a not-for-profit repository, e.g. FigShare, with access details provided in the manuscript.

- 'n' clearly defined (e.g. x cells from y slices in z animals) in the Methods. Authors should be mindful of pseudoreplication.

- All relevant 'n' values must be clearly stated in the main text, figures and tables.
- The most appropriate summary statistic (e.g. mean or median and standard deviation) must be used. Standard Error of the Mean (SEM) alone is not permitted.
- Exact p values must be stated. Authors must not use 'greater than' or 'less than'. Exact p values must be stated to three significant figures even when 'no statistical significance' is claimed.

EDITOR COMMENTS

Reviewing Editor:

Both reviewers acknowledge the significance of this work for the field. Reviewer 2 has raised several concerns that need clarification and further discussion. I agree that revisiting the western blots in Figure 4C is necessary to suggest an increase in p-PKD. Statistics are necessary to determine significance.

Please also see 'Required Items' above.

Senior Editor:

Thank you for contributing this interesting and promising manuscript. It has been scrutinized by two Expert Referees and a Reviewing Editor who agree overall on its potential for high impact on the field. I concur with their insights.

Nonetheless, as you will see in reviewing their detailed comments, there are points require your attention. While the critiques may appear daunting at first glimpse, I anticipate that you will be able to address many points readily, as they largely are points of clarification rather than experimentation (Referee 1 comments; Referee 2, major comments 1 and 2). However, there remain 3 points, raised by Referee 2, that may involve additional experimentation. These are, as follows:

- 1) Please furnish sufficient replicates (or clarify data presentation) for evaluating effect of DCPiB.
- 2) You are encouraged to establish if S1PR2 and/or other S1PRs are expressed (Referee 4, point 4).
- 3) Also, I suggest focusing effort on concerns regarding conclusions drawn from data on Figure 4C (Referee 2, point 5) that are need of additional statistical powering (in other words, likely additional immunoblot analyses) to be fully justifiable.

In your revised manuscript, please do also ensure that presentation of statistics fully complies with The Journal of Physiology's published Statistical Policy. For example, exact p values should be presented to 3 significant figures, as opposed to 3 decimal places. Also note that all errors should be reported as the standard deviation, as opposed to the standard error of the mean.

I thank you for considering The Journal of Physiology and look forward to receiving your revised manuscript.

REFEREE COMMENTS

Referee #1:

In this very interesting manuscript the Stauber group reports that S1P activates VRA in Hela cells using both a FRET assay, which leaves cells unperturbed, on exogenous LRRc8, as well as patch clamp on endogenous VRAC. Exploiting this finding the authors conducted a series of highly convincing experiments to dissect the molecular components of the transduction pathway, allowing the conclusion that Gbetagamma dimers are primarily involved leading to PKD phosphorylation and subsequent VRAC activation. The results are of significant physiological relevance, in particular because osmolarity independent VRAC activation likely plays a role in various physiological setting.

I have only a few minor suggestions.

1. On page 6, the term "raw macrophages" could be misleading for reader not familiar with this cell line.
2. A few lines below: are these breast cell lines, or breast cancer cell lines?
3. In the sentence "We systematically investigate S1P signalling pathway in HeLa cells by sequentially blocking individual steps" I suggest to insert "the" before S1P signaling pathway.
4. On page 9, regarding the sentence "We next tested S1P induction of VRAC currents by whole-cell patch-clamp experiments in HeLa cells, which fully confirmed our FRET data.": I would delete the part "which fully confirmed our FRET data.", because it is not very clear and should be stated after showing the results.
5. On page 13, the last statement of the sentence "To further challenge the notion that LRRC8/VRAC is activated by S1P through G β γ -PLC β signalling, we tested S1P induction of VRAC currents in HEK293 cells, which were reported to express less of the G β γ -sensitive PLC β isoforms PLC β 2 and PLC β 3 compared to HeLa cells." needs a reference.
6. In the discussion, the pioneering experiments of Doroshenko on the activation of VRAC by GTP γ S should be cited. (<https://pubmed.ncbi.nlm.nih.gov/1326043/>, <https://www.ncbi.nlm.nih.gov/pmc/articles/PMC1181530/>).

Referee #2:

In the study entitled "Sphingosine-1-phosphate activates LRRC8 volume-regulated anion channels through G β γ signalling", Kostritskaia et al. attempt to elucidate the mechanisms of VRAC activation by S1P signaling in normotonic condition. The authors leveraged the FRET-based LRRC8/VRAC activation approach they recently developed (PMID: 31210638), in which the C-terminal LRR domain of LRRC8A and LRRC8E are tagged with fluorescent proteins that activate FRET signal by proximity, and the method relies on the assumption that VRAC activation involves the separation of the LRR domains which translates into a reduction of the FRET signal. The authors previously demonstrated that changes in FRET signals are plasma membrane specific and such signals are not affected by LRRC8 subunits that might be accumulated in the ER as a consequence of overexpression of fluorescent-tagged LRRC8 proteins in cells. The authors corroborate their FRET results with patch clamp electrophysiology and then by using agonists and antagonists of the S1P/S1PR/G β γ /PKD the authors propose that such signaling pathway is involved in LRRC8/VRAC activation. The methods the authors used are appropriate and, in most cases, they have included proper controls. In my opinion, this work is worth considering for publication in The Journal of Physiology as the authors are making an attempt to answer one of the open questions in the field of VRACs, which relates to physiological activation mechanisms of VRACs that are distinct from the hypotonic challenges that are rarely found in biological systems. Therefore, the study is relevant to the field and interesting to better understand the physiological role of LRRC8 channels. I will encourage the authors to address some major points which I hope will help to increase the scope of the manuscript and make it suitable for publication in The Journal of Physiology. Please find my comments below:

Major points:

1. Although the main focus of the paper is to elucidate the mechanism of VRAC activation by S1P signaling which translates in Cl⁻ currents, I will encourage the authors to consider including in the discussion the possibility of additional potential transport mechanisms that are uncoupled from channel function (i.e., in the absence of an active signaling pathway that potentially activates and open the channel). There is a significant amount of evidence showing permeation of different substrates by these channels in normotonic conditions in the absence of an active signaling pathway. Therefore, VRACs

very likely have properties of osmolyte transport that are uncouple from channel function similar to other large pore channels such as connexin hemichannels as recently described (Cf. <https://doi.org/10.1101/2024.02.20.581300>).

2. What is the rationale of using 10 nM S1P for FRET experiments and 100 nM for patch clamp and downstream biochemical experiments? The changes in FRET signals are very weak at 10 nM S1P (<10% decay). I am wondering if increasing the S1P concentration 10x (i.e., 100 nM) will improve the FRET signal results? If the authors have a valid justification for using 10 nM for the FRET experiment, please explain. If not, I will suggest increasing the concentration of S1P for FRET experiments. Also, is it possible that the expression of these bulky fluorescent fusion proteins at the C-terminus will pre-activate VRAC and further activation by hypotonic challenge or S1P might not be as sensitive as expected?

3. In Figure 2A, the authors show a significant increase in whole-cell currents traces in isotonic + 100 nM S1P condition, however, such S1P induced currents are partially blocked by the potent VRAC inhibitor DCPIB. DCPIB often blocks completely hypotonic-induced VRAC currents even at lower concentration (i.e. 20 μ M). The authors missed to include statistical analysis and number of repeats for DCPIB experiments. The fact that these data is not quantified and the number of repeats for DCPIB blockade of S1P-induced currents is missing does not help the authors to support their hypothesis. Please add repeats and quantification. Why are not the S1P-induced VRAC currents completely blocked by DCPIB? Are there additional mechanisms (i.e. other chloride channels) that are potentially being activated by S1P? I will encourage the authors to delete LRRC8A expression (as they have nicely done for S1PR1 in subsequent panel 2F,G) and repeat this experiment as this would be the cleanest system to justify these S1P currents are indeed mediated by VRACs. This will be a strong argument for their hypothesis.

4. In order to make a fair conclusion that S1PR1 rather than S1PR2 determines the activation of endogenous LRRC8/VARAC by S1P, the authors must corroborate the expression of other S1PR2 in these cell lines. The fact that by treating the cells with other S1PR antagonists (e.g. JTE-013) show no changes in S1P-triggered VRAC activation doesn't rule out the possibility that they are actually non-functional in the pathway. Indeed, the lack of inhibition could be justified by a lack of expression of additional S1PRs. Therefore, the authors must corroborate, at least by qPCR analysis, the expression of other S1PRs other than S1PR1 to make such a conclusion.

5. The biochemistry data for p-PKD blots shown in Figure 4C,D is, in my honest opinion, not convincing at all. Hypotonic challenge seems to induce an increment in p-PKD as shown in Fig 4A,B but S1P seems it is not. Indeed, for these specific figure panels the authors use S.E.M. and there are overlapping standard errors which can be interpreted as non-significant for a n=3 independent repeats. In fact, the expression of the p-PKD follows the same patterns as the total PKD. But also, the biochemistry data in panel E for the gallein and W123 treatments lack total PKD blots. I will encourage the authors to work on these WBs by increasing the number of repeats as this is an essential component of their signaling pathway and their conclusion.

6. Have the authors measured the resting membrane potential of HeLa cells before and after S1P treatment? What is the rationale for selecting -80 mV for all the quantifications? According to the literature (e.g., doi: 10.1128/iai.64.11.4820-4825.1996), HeLa cells have ~ -50mV RMP. It would be interesting to know if S1P causes changes in membrane potential as this will be relevant to understand the properties of channel activation as the conduction is voltage dependent.

Minor

1. Please use "signaling" instead of "signalling" in the title and throughout the text (it's at least 27 times. Also, in page 6 before (Burow et al., 2015) reference, it is written "purinergic singling" instead of "purinergic signaling", please correct.

2. Please set the scales for HeLa and HEK 293 cells in Fig 5A-C to the same scale. Probably there will be no differences between HeLa and HEK 293 in the iso+S1P condition in 5A and 5B at depolarization step pulses.

END OF COMMENTS

Confidential Review

02-Apr-2024

POINT-BY-POINT RESPONSE TO EDITORS AND REFEREES

EDITOR COMMENTS

Reviewing Editor:

Both reviewers acknowledge the significance of this work for the field. Reviewer 2 has raised several concerns that need clarification and further discussion. I agree that revisiting the western blots in Figure 4C is necessary to suggest an increase in p-PKD. Statistics are necessary to determine significance.

Please also see 'Required Items' above.

> We thank the Reviewing Editor for this assessment. Among other revisions, we have increased the number of independent experiments for the Western blots in Figures 4A and 4C (now Fig. 6) and included the corresponding statistical analysis.

Senior Editor:

Thank you for contributing this interesting and promising manuscript. It has been scrutinized by two Expert Referees and a Reviewing Editor who agree overall on its potential for high impact on the field. I concur with their insights.

Nonetheless, as you will see in reviewing their detailed comments, there are points require your attention. While the critiques may appear daunting at first glimpse, I anticipate that you will be able to address many points readily, as they largely are points of clarification rather than experimentation (Referee 1 comments; Referee 2, major comments 1 and 2). However, there remain 3 points, raised by Referee 2, that may involve additional experimentation. These are, as follows:

- 1) Please furnish sufficient replicates (or clarify data presentation) for evaluating effect of DCPIB.
- 2) You are encouraged to establish if S1PR2 and/or other S1PRs are expressed (Referee 4, point 4).
- 3) Also, I suggest focusing effort on concerns regarding conclusions drawn from data on Figure 4C (Referee 2, point 5) that are need of additional statistical powering (in other words, likely additional immunoblot analyses) to be fully justifiable.

In your revised manuscript, please do also ensure that presentation of statistics fully complies with The Journal of Physiology's published Statistical Policy. For example, exact p values should be presented to 3 significant figures, as opposed to 3 decimal places. Also note that all errors should be reported as the standard deviation, as opposed to the standard error of the mean.

I thank you for considering The Journal of Physiology and look forward to receiving your revised manuscript.

> We thank the Senior Editor for this assessment. As described in our point-by-point response to the Referees' comments below, we have —amongst other revisions—

1) tested different concentrations of DCPIB as well as carbenoxolone and provided the corresponding statistics, showing a clear block of the currents. Additionally, we now show that knock-out of LRRC8A (HeLa cells) or LRRC8A-E (HEK293 cells) prevented the currents;

2) tested for the expression of the five S1PR paralogues;

3) increased the number of independent experiments from three to six for the Western blot data in Figures 4A and 4C (now Figure 6) and including the corresponding statistical analysis, which reveals significant effects.

We have also adjusted the presentation of statistics to comply with the journal's policy.

REFEREE COMMENTS

Referee #1:

In this very interesting manuscript the Stauber group reports that S1P activates VRA in HeLa cells using both a FRET assay, which leaves cells unperturbed, on exogenous LRRC8, as well as patch clamp on endogenous VRAC. Exploiting this finding the authors conducted a series of highly convincing experiments to dissect the molecular components of the transduction pathway, allowing the conclusion that Gbetagamma dimers are primarily involved leading to PKD phosphorylation and subsequent VRAC activation. The results are of significant physiological relevance, in particular because osmolarity independent VRAC activation likely plays a role in various physiological settings.

> We thank the Referee for the evaluation and the positive comments.

I have only a few minor suggestions.

1. On page 6, the term "raw macrophages" could be misleading for reader not familiar with this cell line.

> For clarity, we have changed the term to "macrophage cell line RAW 264.7".

2. A few lines below: are these breast cell lines, or breast cancer cell lines?

> Furuya et al. (2021) reported VRAC activation by S1P for both breast cancer cell lines (MCF7 and MDA-MB231) and a non-carcinogenic breast epithelial cell line (MCF10A). For clarity, we now state "(...) in epithelial breast cancer and non-carcinogenic breast cell lines".

3. In the sentence "We systematically investigate S1P signalling pathway in HeLa cells by sequentially blocking individual steps" I suggest to insert "the" before S1P signaling pathway.

> Corrected as suggested by the referee.

4. On page 9, regarding the sentence "We next tested S1P induction of VRAC currents by whole-cell patch-clamp experiments in HeLa cells, which fully confirmed our FRET data.": I would delete the part "which fully confirmed our FRET data.", because it is not very clear and should be stated after showing the results.

> As suggested by the referee, we deleted the phrase and now state "confirmed our FRET results" after presenting the results.

5. On page 13, the last statement of the sentence "To further challenge the notion that LRRC8/VRAC is activated by S1P through G β -PLC β signalling, we tested S1P induction of VRAC currents in HEK293 cells, which were reported to express less of the G β -sensitive PLC β isoforms PLC β 2 and PLC β 3 compared to HeLa cells." needs a reference.

> We have now cited Lau et al., *Cell Commun Signal* 2013, which was previously only cited in the Discussion.

6. In the discussion, the pioneering experiments of Doroshenko on the activation of VRAC by GTPgammaS should be cited. (<https://pubmed.ncbi.nlm.nih.gov/1326043/>, <https://www.ncbi.nlm.nih.gov/pmc/articles/PMC1181530/>).

> We agree that these works should be cited for the activation of VRAC by GTPyS and have added the references in the first paragraph of the Discussion.

Referee #2:

In the study entitled "Sphingosine-1-phosphate activates LRRC8 volume-regulated anion channels through G β signalling", Kostritskaia et al. attempt to elucidate the mechanisms of VRAC activation by S1P signaling in normotonic condition. The authors leveraged the FRET-based LRRC8/VRAC activation approach they recently developed (PMID: 31210638), in which the C-terminal LRR domain of LRRC8A and LRRC8E are tagged with fluorescent proteins that activate FRET signal by proximity, and the method relies on the assumption that VRAC activation involves the separation of the LRR domains which translates into a reduction of the FRET signal. The authors previously demonstrated that changes in FRET signals are plasma membrane specific and such signals are not affected by LRRC8 subunits that might be accumulated in the ER as a consequence of overexpression of fluorescent-tagged LRRC8 proteins in cells. The authors corroborate their FRET results with patch clamp electrophysiology and then by using agonists and antagonists of the S1P/S1PR/G β /PKD the authors propose that such signaling pathway is involved in LRRC8/VRAC activation. The methods the authors used are appropriate and, in most cases, they have included proper controls. In my opinion, this work is worth considering for publication in *The Journal of Physiology* as the authors are making an attempt to answer one of the open questions in the field of VRACs, which relates to physiological activation mechanisms of VRACs that are distinct from the hypotonic challenges that are rarely found in biological systems. Therefore, the study is relevant to the field and interesting to better understand the physiological role of LRRC8 channels. I will encourage the authors to address some major points which I hope will help to increase the scope of the manuscript and make it suitable for publication in *The Journal of Physiology*. Please find my comments below:

> We thank the reviewer for the thorough review and the constructive criticism, which we have

addressed as shown below. This has greatly strengthened our manuscript.

Major points:

1. Although the main focus of the paper is to elucidate the mechanism of VRAC activation by S1P signaling which translates in Cl⁻ currents, I will encourage the authors to consider including in the discussion the possibility of additional potential transport mechanisms that are uncoupled from channel function (i.e., in the absence of an active signaling pathway that potentially activates and open the channel). There is a significant amount of evidence showing permeation of different substrates by these channels in normotonic conditions in the absence of an active signaling pathway. Therefore, VRACs very likely have properties of osmolyte transport that are uncouple from channel function similar to other large pore channels such as connexin hemichannels as recently described (Cf. <https://doi.org/10.1101/2024.02.20.581300>).

> We thank the referee for the suggestion to add this interesting point to our Discussion. Although most studies on LRRC8 large-pore channels find that the activation of channel function (electrophysiological currents) and the transport of various osmolytes (such as ATP) are similarly induced by hypotonicity or isotonic cues such as S1P, an uncoupling of osmolyte transport from channel might be possible, as recently shown for pannexin channels (Gaete et al., *PNAS* 2024). We have now added a corresponding paragraph to the Discussion.

2. What is the rationale of using 10 nM S1P for FRET experiments and 100 nM for patch clamp and downstream biochemical experiments? The changes in FRET signals are very weak at 10 nM S1P (<10% decay). I am wondering if increasing the S1P concentration 10x (i.e., 100 nM) will improve the FRET signal results? If the authors have a valid justification for using 10 nM for the FRET experiment, please explain. If not, I will suggest increasing the concentration of S1P for FRET experiments. Also, is it possible that the expression of these bulky fluorescent fusion proteins at the C-terminus will pre-activate VRAC and further activation by hypotonic challenge or S1P might not be as sensitive as expected?

> In different studies, various concentrations of S1P have been applied to activate VRAC. The first report on the activation of VRAC by S1P (Burow et al., *Pfl Arch* 2015) used S1P at a concentration of 10 nM, which was also employed subsequently by the same group (Zahiri et al, *BBA* 2021) and others (Chu et al., *Sci Adv* 2021). The decrease in FRET intensity by <10 % is weaker but in the same range as that induced by ~25% hypotonicity (Fig. 1B and König et al., *Elife* 2019). We used 100 nM S1P to study signal transduction by electrophysiology, as this concentration was previously used by Furuja et al. (*Life* 2021) who reported an EC50 of 37.5 nM for S1P-induced ATP release. We have also performed whole-cell patch-clamp measurements using 10 nM S1P (now shown in Fig. 2C) and found no significant differences in current densities compared to 100 nM S1P. Therefore, the weaker drop in FRET upon S1P application compared to hypotonicity is in agreement with the currents induced by 100 nM S1P being smaller than those induced by 25% hypotonicity (Fig. 7).

The potential basal activation of LRRC8 channels by the fusion of fluorescent proteins at their C-termini (as shown by Gaitán-Peñas et al., *Biophys J* 2016) is indeed a concern regarding the optical sensor in general, as discussed in our previous publications (König et al., *Elife* 2019; Klüssendorf et al., *JoVE* 2024). Despite a potentially reduced sensitivity (which is, of course, hypothetical because there would be no sensor without the fusion), the FRET assay has been shown to robustly monitor the activity of LRRC8/VRAC. Furthermore, in this study, the data are consistent with the electrophysiological measurements.

3. In Figure 2A, the authors show a significant increase in whole-cell currents traces in isotonic + 100 nM S1P condition, however, such S1P induced currents are partially blocked by the potent VRAC inhibitor DCPIB. DCPIB often blocks completely hypotonic-induced VRAC currents even at lower concentration (i.e. 20 μ M). The authors missed to include statistical analysis and number of repeats for DCPIB experiments. The fact that these data is not quantified and the number of repeats for DCPIB blockade of S1P-induced currents is missing does not help the authors to support their hypothesis. Please add repeats and quantification. Why are not the S1P-induced VRAC currents completely blocked by DCPIB? Are there additional mechanisms (i.e. other chloride channels) that are potentially being activated by S1P? I will encourage the authors to delete LRRC8A expression (as they have nicely done for S1PR1 in subsequent panel 2F,G) and repeat this experiment as this would be the cleanest system to justify these S1P currents are indeed mediated by VRACs. This will be a strong argument for their hypothesis.

> We have now included the statistical analysis for DCPIB blockade of S1P-induced currents (Fig. 2C) and replaced the current traces with a more representative example (Fig. 2A). Additionally, we tested lower concentrations of DCPIB (20 μ M) and another VRAC inhibitor, carbenoxolone, which completely block S1P-induced currents (Fig. 2C).

Moreover, we confirmed that the S1P-induced currents are mediated by LRRC8/VRAC by using LRRC8-depleted cells, showing absence of these currents in LRRC8A-depleted HeLa cells obtained from the Qiu lab (Fig. 2C and D). Furthermore, while overexpression of PLCbeta3 facilitated S1P-induced currents in HEK293 cells (now Fig. 7C), it did not have this effect in LRRC8-depleted HEK293 cells obtained from the Jentsch lab (Fig. 7D).

4. In order to make a fair conclusion that S1PR1 rather than S1PR2 determines the activation of endogenous LRRC8/VRAC by S1P, the authors must corroborate the expression of other S1PR2 in these cell lines. The fact that by treating the cells with other S1PR antagonists (e.g. JTE-013) show no changes in S1P-triggered VRAC activation doesn't rule out the possibility that they are actually non-functional in the pathway. Indeed, the lack of inhibition could be justified by a lack of expression of additional S1PRs. Therefore, the authors must corroborate, at least by qPCR analysis, the expression of other S1PRs other than S1PR1 to make such a conclusion.

> We have tested for the expression of the S1PR1 paralogues by RT-PCR (Fig. 3C). Although not quantitatively (in qRT-PCR all five had low CT values, not shown), we found all five paralogues expressed at mRNA level (we unsuccessfully tried several commercial antibodies against S1PR2 in immunoblot). However, S1PR1 seems to be predominantly involved in S1P-induced VRAC activation, as the application of the S1PR1 antagonist W123 and W146 or knock-out of S1PR1 strongly diminished S1P-induced currents.

Following up on the referee's suggestion that we cannot exclude a role for S1PR2 in S1P-triggered LRRC8/VRAC activation, even though its pharmacological inhibition did not significantly reduce S1P-induced currents, we also tested whether its overexpression could rescue S1P-induced currents in S1PR1-deficient cells. Indeed, we observed S1P-induced currents not only upon re-expression of S1PR1 but also upon overexpression of S1PR2 (Fig. 4).

Together, this shows that S1PR2 is capable of mediating S1P-triggered VRAC activation, but at endogenous expression levels, S1PR1 predominantly drives the activation of LRRC8/VRAC by S1P. We have updated the subtitle in the Results and adjusted the Discussion accordingly.

5. The biochemistry data for p-PKD blots shown in Figure 4C,D is, in my honest opinion, not convincing at all. Hypotonic challenge seems to induce an increment in p-PKD as shown in Fig 4A,B but S1P seems it is not. Indeed, for these specific figure panels the authors use S.E.M. and there are overlapping standard errors which can be interpreted as non-significant for a n=3 independent repeats. In fact, the expression of the p-PKD follows the same patterns as the total PKD. But also, the biochemistry data in panel E for the gallein and W123 treatments lack total PKD blots. I will encourage the authors to work on these WBs by increasing the number of repeats as this is an essential component of their signaling pathway and their conclusion.

> As correctly suggested by the reviewer, we have increased the number of replicates for hypotonicity- and S1P-induced PKD activation (to six independent experiments) and added the total-PKD blots for gallein and W123 treatments. Moreover, we now show the ratio of p-PKD to total PKD and present the data along with their mean and standard deviation (S.D.), as well as the statistical analysis. The conclusion from these results (now Fig. 6) remains unchanged.

6. Have the authors measured the resting membrane potential of HeLa cells before and after S1P treatment? What is the rationale for selecting -80 mV for all the quantifications? According to the literature (e.g., doi: 10.1128/iai.64.11.4820-4825.1996), HeLa cells have ~ -50mV RMP. It would be interesting to know if S1P causes changes in membrane potential as this will be relevant to understand the properties of channel activation as the conduction is voltage dependent.

> We have not measured the resting membrane potential. However, it is known that the activation of LRRC8 anion channels shifts the membrane potential towards the equilibrium potential of chloride, which in turn will affect other ion channels and alter the driving force for other ions. This contributes, e.g., to glucose-stimulated insulin secretion in pancreatic β -cells (Kang et al., Nat Commun 2018; Stuhlmann et al., Nat Commun 2018). The moderate outward rectification of LRRC8/VRAC channels is even observed for single-channel conductance (Gaitán-Peñas et al., Biophys J 2016). Therefore, while changes in membrane potential within the inside-negative voltage range will affect the driving force for chloride, they will likely have little impact on LRRC8/VRAC open probability.

We selected -80 mV for the quantification because it is widely used in the literature (including in our previous studies) for quantification or as a holding potential in electrophysiological measurements of LRRC8/VRAC, independent of cell type, including HeLa cells. Given that the equilibrium potential for chloride in our recording solutions is approximately -30 mV, chloride currents through open LRRC/VRAC channels at -50 mV would be qualitatively similar to those at -80 mV, albeit with lower amplitude due to the reduced driving force for chloride ions.

Minor

1. Please use "signaling" instead of "signalling" in the title and throughout the text (it's at least 27 times). Also, in page 6 before (Burow et al., 2015) reference, it is written "purinergic singling" instead of "purinergic signaling", please correct.

> Since the journal's Information for Authors states that UK English spelling and terminology should be used, we have retained the spelling of "signalling". We thank the reviewer for spotting the typo "purinergic singling" and have corrected it.

2. Please set the scales for HeLa and HEK 293 cells in Fig 5A-C to the same scale. Probably there will be no differences between HeLa and HEK 293 in the iso+S1P condition in 5A and 5B at depolarization step pulses.

> We now show the current traces using the same scale (now Fig. 7). In the depicted examples, S1P-induced currents in the (unphysiological) inside-positive voltage range are only moderately larger in the HeLa cell than in the HEK293 cell. Importantly, in the negative voltage range, HEK293 gave significantly smaller currents than HeLa cells (Fig. 7E).

Dear Dr Stauber,

Re: JP-RP-2024-286665R1 "Sphingosine-1-phosphate activates LRRc8 volume-regulated anion channels through G β y signalling" by Yulia Kostrikskaia, Sumaira Pervaiz, Anna Klemmer, Malte Klüssendorf, and Tobias Stauber

We are pleased to tell you that your paper has been accepted for publication in The Journal of Physiology.

Authors should note that it is too late at this point to offer corrections prior to proofing. Major corrections at proof stage, such as changes to figures, will be referred to the Editors for approval before they can be incorporated. Only minor changes, such as to style and consistency, should be made at proof stage. Changes that need to be made after proof stage will usually require a formal correction notice.

If you would like to receive our 'Research Roundup', a monthly newsletter highlighting the cutting-edge research published in The Physiological Society's family of journals (The Journal of Physiology, Experimental Physiology and Physiological Reports), please click this link, fill in your name and email address and select 'Research Roundup': <https://www.physoc.org/journals-and-media/membernews/>.

Yours sincerely,

Peying Fong
Senior Editor
The Journal of Physiology

P.S. - You can help your research get the attention it deserves! Check out Wiley's free Promotion Guide for best-practice recommendations for promoting your work at www.wileyauthors.com/eeo/guide. You can learn more about Wiley Editing Services which offers professional video, design, and writing services to create shareable video abstracts, infographics, conference posters, lay summaries, and research news stories for your research at www.wileyauthors.com/eeo/promotion.

IMPORTANT NOTICE ABOUT OPEN ACCESS: To assist authors whose funding agencies mandate public access to published research findings sooner than 12 months after publication, The Journal of Physiology allows authors to pay an Open Access (OA) fee to have their papers made freely available immediately on publication.

You can check if your funder or institution has a Wiley Open Access Account here: <https://authorservices.wiley.com/author-resources/Journal-Authors/licensing-and-open-access/open-access/author-compliance-tool.html>.

EDITOR COMMENTS

Reviewing Editor:

I agree with the revisions.

Senior Editor:

Your revised manuscript, "Sphingosine-1-phosphate activates LRRC8 volume-regulated anion channels through G β γ signalling", has been evaluated by the original Referees. They express satisfaction with how this present version incorporates feedback on the initially submitted manuscript, as does the Reviewing Editor. Overall, we agree that this newly identified signalling pathway for LRRC8 channel activation holds high impact for the field at large. Congratulations!

We thank you for submitting your manuscript for consideration by The Journal of Physiology.

REFEREE COMMENTS

Referee #1:

The authors have addressed all concerns in a satisfactory manner.

Referee #2:

The authors have satisfactorily addressed all my concerns. I have nothing more to add than to congratulate the author for the impressive work. The revised version is much stronger than the original submitted article.

1st Confidential Review

30-Sep-2024